# Synthetic Theorem Generation in Lean

## Abstract

The application of large language models (LLMs) to theorem proving presents a promising avenue for advancing formal mathematics. Interactive theorem provers, such as Lean, offer a rigorous framework within which these models can assist in or automate proof discovery, grounding their reasoning capabilities in a sound, verifiable formal system. However, the potential of LLMs in this domain is constrained by the limited availability of formal proof corpora for training. To address this limitation, we introduce a synthetic theorem generator capable of producing novel Lean theorems and their corresponding proofs. Our approach employs forward reasoning to synthesize new propositions from premises drawn from existing Lean libraries. We explore candidate reasoning steps using a search strategy that optimizes for diversity of output, apply them in a linear fashion that avoids irrelevant proof steps, and assess their effect by metaprogrammatically executing corresponding Lean tactics. These methods enable the generation of an arbitrary number of new theorems and proofs across various mathematical domains, using common Lean proof tactics while ensuring the correctness of generated theorems by construction. We demonstrate the efficacy of the generated theorems and training data by fine-tuning models on synthetic theorems and evaluating them on the miniF2F-test benchmark. Our results show improvements in theorem-proving capabilities, with accuracy increasing from 37.3% to 38.5% for the Falcon2-11B model trained solely on Mathlib, and from 38.1% to 39.3% for the same model trained on a mix of rich datasets. These improvements highlight the value of our diverse synthetic data in augmenting limited existing corpora of formal proofs, providing complementary information that enhances LLMs' performance on theorem-proving tasks even when combined with other datasets.

## 1 Introduction

Interactive theorem proving (ITP) provides a foundation for mechanically certifiable formal mathematics and software verification. While many proof assistants, such as the Lean theorem prover (de Moura & Ullrich, 2021), offer tools to automate portions of common proof-writing tasks, formal proofs still frequently require considerable time, effort, and expertise. One exciting direction of research within ITP aims to address this issue by using large language models (LLMs) to assist in or automate the writing of formal proofs in languages like Lean (Yang et al., 2023). LLMs have proven useful for code generation (Chen et al., 2021), though they continue to exhibit hallucinations in code-writing tasks by producing subtle bugs (Ji et al., 2023). Unlike code generated in languages such as Python or C, however, the correctness of formal proofs can be automatically verified using proof assistants.

The use of LLMs for ITP depends upon the availability of large quantities of formal proofs from which to extract training data. Yet existing corpora of formalized mathematics remain relatively scarce. Formal theorem proving is often time-intensive and requires specialized knowledge of both the relevant mathematical domains and the tools used for formalization, limiting the rate at which formalized mathematics is produced. As of this writing, the Lean theorem prover's mathematical library, Mathlib, is the primary existing corpus of Lean code, containing approximately 1.5 million lines of code and growing by just over 300,000 lines in the past year (The mathlib Community, 2024). Despite the sizable portion of modern mathematics contained in this library, it is significantly smaller than corpora available in other languages, which total hundreds of millions of lines (Chen et al., 2021).

To obtain sufficient data for training at large scales, therefore, it is often necessary to synthetically generated data. Prior techniques for generating synthetic data in formal languages have been varied. Some use random sampling or LLM-based methods to generate conjectures, for which proofs are then generated using proof search (An et al., 2024; Xin et al., 2024; Ying et al., 2024; Zombori et al., 2021). Other techniques procedurally generate new theorems and their proofs simultaneously by successively applying inference rules to reason forward from existing theorems (Firoiu et al., 2021; Lample et al., 2022; Polu & Sutskever, 2020; Trinh et al., 2024; Wang & Deng, 2020).

We introduce a synthetic data generator for Lean based on a procedural forward-reasoning approach and distinguished by several key features. First, our initial hypotheses are drawn from proofs in existing Lean libraries, and our inference steps use LLM-based premise selection to identify relevant lemmas in Mathlib, resulting in theorems that reference definitions of interest in modern mathematics. Second, we generate proofs by applying commonly-used Lean tactics, producing proofs with reasoning steps similar to human-written proofs. Third, we ensure the diversity and quality of our generator's output by using a search procedure optimized to produce dissimilar theorems and a linear generation strategy that precludes irrelevant proof steps.

## 2 BACKGROUND

Lean 4 is a dependently typed functional programming language and theorem prover based on the Calculus of Inductive Constructions (de Moura & Ullrich, 2021). It has been used to formalize a wide array of mathematics, notably in the community-driven Mathlib project (The mathlib Community, 2020), which contains formalizations of over 150,000 theorems across various mathematical fields (The mathlib Community, 2024).

Many proofs in Lean are written using Lean's metaprogrammatic *tactic* system. Instead of providing an explicit proof term, users may specify a series of tactics that correspond to high-level reasoning steps. These tactics generate the underlying proof term that is checked by Lean. An annotated example of a tactic proof in Lean is shown in Appendix I.

Theorem proving in Lean is interactive. After each tactic step, Lean displays to the user the current goal and a list of *hypotheses* that have been added to the local context. By "hypotheses," we mean not only the antecedents of the theorem statement that are assumed to be true within the proof, but also any propositions derived therefrom. Collectively, we refer to the goal and hypotheses as the *proof state*. Users can incrementally develop a proof, observing the effect of each tactic on the proof state in real time and receiving immediate feedback if a tactic does not succeed. Because the tactic system offers this flexibility and interactivity, and because of its ubiquity as a mode of interaction with the proof assistant, our work aims specifically to develop training data for tactic-based theorem proving.

## 3 RELATED WORK

### 3.1 SYNTHETIC FORMAL-PROOF GENERATION

Several techniques have previously been employed to generate synthetic data in Lean and other formal languages. Some approaches generate theorem statements independently of their proofs. Several such approaches use random sampling within a fixed domain—such as integer arithmetic or propositional logic—to generate known-true theorem statements (An et al., 2024; Zombori et al., 2021). Others use autoformalization: theorem statements in natural language are converted to formal statements by an LLM (Xin et al., 2024; Ying et al., 2024). In both cases, a separate proof search procedure is required to generate the corresponding proof. Moreover, when using techniques like autoformalization, it is possible that the proposed theorem statement is incorrect; Xin et al. (2024) address this eventuality by searching for proofs of both the proposed theorem statement and its negation.

Other data-generation techniques procedurally generate both new theorems and their proofs simultaneously through forward reasoning (Firoiu et al., 2021; Lample et al., 2022; Polu & Sutskever, 2020; Trinh et al., 2024; Wang & Deng, 2020). These techniques iteratively apply inference rules to existing theorems, the result of which is a theorem whose proof consists of the applied inference

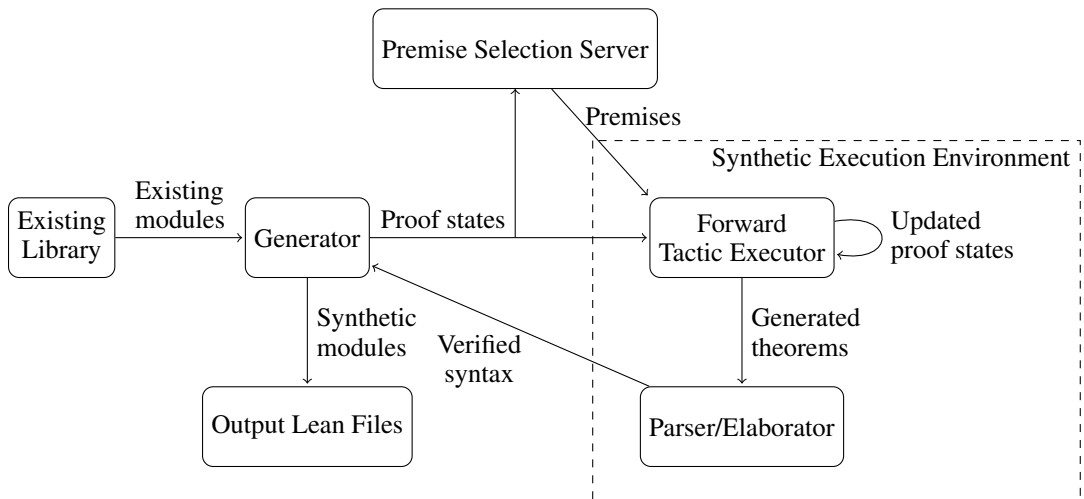

Figure 1: An overview of our synthetic theorem generator's architecture. Starting from proof states extracted from an existing library, forward-reasoning tactics are iteratively executed, yielding theorems and corresponding tactic proofs that are exported as Lean modules.

rules. Most such techniques involve either random sampling or exhaustion of available proof steps, though Wang & Deng (2020) train neural networks to identify desirable proof steps.

## 3.2 LLMs FOR LEAN

LLMs have been applied to a variety of tasks related to theorem-proving in Lean. Han et al. (2022) train language models, using data extracted from both tactic commands and raw proof terms, on a range of tasks including predicting the next lemma to be applied in a proof and the types of partial and complete proof terms, demonstrating the importance of a rich dataset from which a variety of term- and tactic-level data can be extracted. More recently, as part of the LeanDojo project, Yang et al. (2023) introduced the ReProver model, a retrieval-augmented tactic generator for Lean. ReProver incorporates a Dense Passage Retriever-based (Karpukhin et al., 2020) premise retriever, which predicts library lemmas that may be relevant to the next step of a proof. While ReProver uses these predictions to enhance tactic prediction, premise selection is also applicable to the task of synthetic theorem generation; we discuss our generator's use of ReProver in Section 4.4. Additionally, Xin et al. (2024) demonstrate the capacity of synthetic data to improve the performance of the DeepSeekMath 7B model on Lean theorem-proving tasks, though their synthetic data, unlike ours, was produced using autoformalization of theorem statements and proof search.

## 4 APPROACH

### 4.1 FORWARD REASONING WITH LEAN TACTICS

Like the other procedural approaches we discuss in Section 3, ours is based on *forward reasoning*. That is, we apply inference rules to existing hypotheses to conclude new ones. For example, from the hypotheses $x \leq y$ and $y \leq z$, we can use the transitivity of the $\leq$ relation to conclude $x \leq z$. Alternatively, we could apply the additivity of $\leq$ to these same hypotheses to conclude $x+y \leq y+z$. This differs from backward reasoning, which reduces a predetermined goal to sufficient subgoals. Reasoning backward, we begin with the goal of proving $x \leq z$ and use the transitivity of $\leq$ to reduce this to proving $x \leq y$ and $y \leq z$. As these examples demonstrate, forward reasoning, unlike backward reasoning, does not require *a priori* knowledge of the statement one aims to prove. This allows a forward reasoning-based search procedure to freely explore a variety of theorems to generate.

We opt for a forward-reasoning approach because of its efficiency, scalability, and adjustability. Unlike approaches that separately generate conjectures and their proofs, forward reasoning combines

theorem generation and proof search into a single procedure. This avoids proof-search failures and precludes the generation of invalid conjectures by ensuring that all generated theorem statements are correct by construction. Additionally, unlike approaches based on auto-formalization, a forward reasoning-based approach like ours does not require access to an input corpus of natural-language mathematics. Lastly, forward reasoning can be tailored to produce proofs of a certain length or of a certain kind by adjusting the number and type of forward-reasoning steps taken during the search procedure.

We differentiate our approach to forward reasoning in several respects. As previously noted, we implement our forward reasoning-based theorem generator in Lean, which features a more complex proof system than those in which similar generators have been previously implemented, such as Metamath (Wang & Deng, 2020). Additionally, we select the initial hypotheses for our forward proofs by drawing on the hypotheses that arise in existing Lean proofs. Finally, rather than applying individual inference rules of the underlying logic, we use Lean tactics that more closely replicate the types of reasoning steps used in human-written proofs. An overview of our architecture is shown in Figure 1.

Forward reasoning depends upon access to an initial collection of hypotheses from which to reason. We obtain these by extracting the current proof state—which includes all available hypotheses at the current point in the proof—from each step of tactic proofs in an existing library of Lean proofs, such as Mathlib or Lean Workbook (Ying et al., 2024). Each extracted state is then used as the initial proof state for proof synthesis, providing our pool of initial hypotheses.

We choose to sample from existing proof states in this manner for several reasons. First, because of the breadth of libraries like Mathlib and Lean Workbook, this sampling allows us to generate a large number of theorems across a wide array of mathematical disciplines. Second, because these libraries comprise theorems of mathematical interest, their hypotheses are likely to entail mathematically interesting propositions and are unlikely to be inconsistent, which would yield trivial theorems. Finally, as we discuss in Section 4.4, because both libraries are based on definitions in Mathlib, we are able to employ Mathlib's sizable collection of lemmas in our forward reasoning.

Once the generator has extracted a proof state from an existing library, it applies a sequence of forward-reasoning tactics to derive a new hypothesis that follows from the available ones. It continues applying tactics until none succeed or a user-determined maximum proof length is reached. Using Lean tactics allows us to produce proofs that better resemble human-written proofs, rather than, for instance, raw proof terms that can be difficult to read and do not resemble most medium- or large-scale proofs written by human authors.

This approach is based on a method of metaprogrammatic tactic execution, simulating the process of writing tactics as a user of Lean. When writing a proof in Lean, the current proof state is visible in a panel in the user's code editor; as the user enters new tactics into the open file, the proof state updates to reflect the changes made by each tactic step. We simulate this process by syntactically constructing tactics that correspond to candidate forward-reasoning steps, then executing these tactics in a metaprogrammatic environment that replicates Lean's in-editor tactic execution. After each tactic step, we assess whether the invoked tactic successfully introduced a new hypothesis into the proof context. If so, we use the new state as the initial state for a new search; otherwise, we discard it. Because our generator is directly executing tactic code, it can produce a tactic proof equivalent to the reasoning carried out during the generation procedure simply by concatenating the tactics executed at each step.

Finally, once the generator has derived a final hypothesis $p$ from the hypotheses $h_1, \ldots, h_n$ in the initial extracted proof state, it outputs the corresponding theorem $h_1 \rightarrow \cdots \rightarrow h_n \rightarrow p$. The proof of this theorem, as noted above, is obtained from the forward-reasoning steps used to derive $p$. Figure 2 demonstrates how a theorem is synthesized in this manner. In the following sections, we describe in greater detail the process of forward proof synthesis and theorem output.

## 4.2 Proof Synthesis Algorithm

To generate these proofs, we attempt to apply all possible tactic steps from an inventory (detailed in Section 4.3) of allowable forward-reasoning tactics. For every successful tactic application, we recursively apply the same search procedure to the resulting proof state. This leads to a potentially

```
theorem mod_eq_zero {a b : R} : a %
    b = 0 ↔ b | a :=
⟨fun h => by ...,
  ... ⟩
```

(1) Mathlib Module Code

```
R : Type u_1
inst† : EuclideanDomain R
a b : R
h : a % b = 0
⊢ b | a
```

(2) Initial Proof State

Extract proof state

Apply forward-reasoning tactics:
```
simp_arith at h
have aux_1 : b * b | b * a :=
    mul_dvd_mul_left b h
ring_nf at aux_1
```

```
theorem ex_0 {R : Type u}
    [EuclideanDomain R]
    {a : R} {b : R}
    (h : a % b = 0) :
    b ^ 2 | b * a :=
by
simp_arith at h
have aux_1 : b * b | b * a :=
    mul_dvd_mul_left b h
ring_nf at aux_1
exact aux_1
```

(4) Synthesized Theorem & Proof

Assemble new theorem statement and proof

```
R : Type u_1
inst† : EuclideanDomain R
a b : R
h : b | a
aux_1 : b ^ 2 | b * a
⊢ b | a
```

(3) Final Proof State

Figure 2: An example of the process by which a theorem is synthesized starting from a Mathlib proof state. Note that the original goal $b \mid a$ is ignored during forward reasoning, which operates only on hypotheses. See Appendix F for a more detailed analysis of the same example.

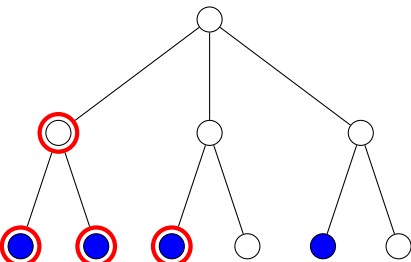

Figure 3: A comparison of naïve search and our algorithm assuming a minimum proof length of 1, maximum proof length of 2, and truncating after four theorems. Nodes in the tree correspond to proof states, and edges to successful tactic applications. Nodes visited by a naïve search are circled in red; those visited by our procedure are shaded blue. As shown, naïve post-order traversal yields proofs with significant overlap and leaves the rightmost subtree unexplored; our procedure produces proofs beginning with each possible initial tactic before re-visiting any subtree.

exponential increase in candidate proofs. Accordingly, when setting the desired number of tactics to higher values, it is often desirable to truncate the search prior to exhausting the search space.

However, using naïve depth-first search, such truncation leads to highly homogeneous output theorems. Since we prefer to generate proofs with as many tactic applications as possible up to the user-specified depth limit, a naïve depth-first approach performs a post-order traversal of the search tree (skipping nodes that do not meet a user-specified minimum depth). Truncating such a traversal yields nodes in tightly clustered subtrees of the search space—that is, proofs with the same initial subsequence of tactics that diverge only near the end of the proof. This is undesirable, as it limits the diversity of our synthetic proofs.

Instead of DFS, therefore, we employ a search algorithm that is optimized to find nodes with distant common ancestors—that is, proofs that diverge from one another as early as possible. Our algorithm iteratively performs a series of depth-first searches to find nodes at the desired depth; however, we start each search from the unexplored children of the shallowest remaining node. This allows us to traverse unexplored subtrees whose common ancestor with any we have already explored is as shallow as possible. Thus, our approach optimizes for finding theorems of greater length as well as

for disjointness among generated theorems. Accordingly, unlike with DFS, a truncated search using our algorithm still yields diverse proof output, making it preferable for high-depth proof generation. A graphical depiction of our algorithm's optimization is shown in Figure 3, and we include the pseudocode for our algorithm in Appendix B.

We include several additional optimizations in our search procedure to enhance both the diversity of our synthetic dataset and the performance of the generator. To promote a diverse dataset, we provide an option to prevent the generation of multiple proofs of the same theorem, which, given the limited inventory of tactics with which the generator operates, are likely to be similar to one another. We optionally apply deduplication to hypotheses as well, preventing the generation of multiple hypotheses corresponding to the same proposition, which could also lead to similar theorems as well as unnecessarily roundabout proofs. We also provide an option to prevent repeated applications of the same library lemma, which can lead to large, repetitive theorem statements. Additionally, to improve performance, we cache unsuccessful proof steps to avoid attempting to invoke tactics on hypotheses of a type to which they have previously failed to apply.

As our proof synthesis procedure proves new propositions, the generator can either output each individually as a distinct theorem or conjoin multiple propositions into a multi-part theorem statement. In the latter case, the proofs of each conjunct appear as discrete subproofs of the larger proof of the conjoined theorem statement. These examples compartmentalize complex proofs into distinct subtasks, demonstrating principles of higher-level proof organization and decomposition to an LLM trained on these data.

## 4.3 Linear Tactic Selection

In order to perform this search procedure, we must have access to an inventory of tactics capable of reasoning forward from a given set of hypotheses. While many Mathlib tactics involve backward reasoning, a small subset are capable of operating strictly forward. We select several of these tactics to use in our forward generation procedure. These include:

- `have`, which introduces a new hypothesis using a lemma from the library;

- `rewrite`, which performs substitution of equal expressions in a given hypothesis;

- `simp_arith`, which simplifies a hypothesis using reductions, library-provided "simplification lemmas," and arithmetic identities; and

- Normalization tactics for expressions involving ring operations (`ring_nf`) and numerals (`norm_num1`).

However, the tactic-execution procedure described in Section 4.1 is relatively agnostic to the specific tactics employed. Therefore, our generator is extensible with additional forward-reasoning tactics, which could be explored in future work to increase both the quantity and diversity of generated proofs.

During an iteration of our generation procedure, we attempt to apply every tactic above to each of the hypotheses in the current proof state. Each successful tactic application generates a new proof state to which we can apply additional tactics. If we have already applied the user-defined maximum number of tactics in a proof, or if we have applied at least the user-defined minimum number of tactics and no further tactics have succeeded in the current proof state, we cease our search from that proof state and output the theorem most recently yielded by our forward reasoning.

It is necessary, however, to restrict our candidate tactics so that the resulting proof does not contain unnecessary or irrelevant steps. This is because arbitrary tactic applications may yield proof steps that have no bearing on the chain of inferences used to arrive at the final theorem statement. For instance, a proof step might add a hypothesis to our context that we do not reference again for the remainder of the proof. Doing so would reduce the quality of training data obtained from our generator, as such steps are undesirable in and irrelevant to the proof of the resulting theorem statement. Therefore, we generate our tactic steps using a linear procedure, similar to the linear resolution scheme employed by Firoiu et al. (2021), that precludes such irrelevant steps. Specifically, we require that each tactic (after the first) make use of the hypothesis introduced or modified by the preceding tactic (though it may additionally use others). This ensures that every tactic we apply

```
theorem ex (a b : ℕ) (hab : a + 2 = b + 5)
         (ha : 10 ≤ a) (ha' : a ≤ 50) : b ≤ 47 := by
   simp_arith at hab    -- hab : a = b + 3
   rewrite [hab] at ha'  -- ha' : b + 3 ≤ 50
   rewrite [hab] at ha   -- ha : 10 ≤ b + 3
   simp_arith at ha'    -- ha' : b ≤ 47
   exact ha'
```

Figure 4: An irrelevant tactic step, highlighted in yellow. The hypothesis `ha`, which this step rewrites, is not used in the final theorem. The actual proof, which omits the highlighted line, proceeds linearly by using the last-modified hypothesis at each step, as depicted by the arrows.

introduces or modifies a hypothesis that is used to arrive at our final conclusion. See Figure 4 for an example.

Lastly, we provide an option to enable *proof minimization* to reduce the likelihood of generating unnecessarily long proofs. Lean possesses several powerful backward-reasoning tactics that can sometimes elide many steps—or, occasionally, entire proofs—performed using the forward-reasoning tactics our generator employs. Such tactics include the `omega` tactic, which can prove equalities and inequalities of natural-number- and integer-valued expressions, and Aesop (Limperg & From, 2023), a general proof-search tactic. However, because these tactics are based on backward reasoning, they cannot be used as part of our forward generation procedure. Instead, we apply minimization as a post-processing step: after generating a forward proof, we attempt to replace a maximal terminal subsequence of the forward-reasoning tactics used in the original proof with a "finishing" tactic like `omega` or Aesop. If such a substitution succeeds, we output this shorter proof rather than the original, strictly forward one. These minimized proofs diversify the tactics appearing in our output proofs by demonstrating instances in which higher-level tactics can be used to quickly discharge a goal. They also produce training data that promotes the use of powerful proof automation in place of longer and potentially roundabout forward proofs. Nonetheless, we leave it as an optional setting due to the increased performance cost of checking proofs for minimizability as well as the potential for circumstances in which forward, more explicit proofs might be preferable to efficient but less verbose ones.

## 4.4 PREMISE SELECTION INTEGRATION

An important step in generating forward proofs is the selection of lemmas (or *premises*) that can be used to construct new hypotheses using the `have` tactic. We draw these premises from Lean's mathematical library, Mathlib. However, given the large number of theorems in Mathlib, it is impractical to attempt to use all or even a significant fraction of those that are available. Instead, we select a limited pool of premises from Mathlib for use in a given round of theorem generation.

To ensure that we select a pool of premises that are relevant to the in-context data and hypotheses, we use the LeanDojo ReProver premise-selection model. This model is capable of identifying premises in Mathlib that are relevant for proving the current goal given the current proof state (Yang et al., 2023). During theorem generation, each initial proof state is passed to ReProver, which produces a list of relevant premises. These premises are then given as input to our tactic-generation procedure alongside the initial proof state. To add diversity to our pool of premises, we also allow some premises to be drawn at random from Mathlib, replacing a specified proportion of the model-selected premises from ReProver. The ratio of random to model-selected premises is a parameter of our generator, and we report its performance with various values of this parameter in Section 5.1.

To improve performance, premise selection runs as a server process distinct from the main generator. This modularity allows multiple generation jobs to run in parallel while maintaining only a single instance of the ReProver model in memory, and it facilitates the separation of the CPU-intensive generation pipeline from the GPU-intensive premise-selection task. Moreover, this modular architecture enables the potential substitution of other premise-selection models in place of ReProver. For instance, a model explicitly calibrated to the task of identifying premises capable of reasoning forward from hypotheses in a given proof state—rather than, like ReProver, selecting premises relevant to closing the proof state's goal—is an avenue for future work that could improve the quality of premise selection in our generator.

Once we have selected an initial pool of premises, we repeatedly sample a random subset of this pool to attempt to apply at each stage of our search. This allows us to draw on a wider range of premises in the course of our search than would repeatedly attempting to apply the same collection of premises at every step. When attempting to apply a premise during generation, we exhaustively search the context for type-correct arguments to which the premise can be applied. We also attempt to synthesize any type-class instance arguments required by the premise by invoking Lean's built-in type-class resolution. If a premise can be applied successfully to yield a new hypothesis, we produce a corresponding `have` tactic that adds the identified hypothesis to the context. Because we must construct these premise applications prior to generating the corresponding tactic syntax, `have` is the one tactic we do not directly metaprogrammatically execute; instead, we directly inject the synthesized hypothesis into the context.

## 4.5 VERIFICATION AND ENVIRONMENT RECONSTRUCTION

Once the generator has constructed a proof, the resulting theorem is written to a Lean file. The theorem statement is the type of the final hypothesis produced by our forward reasoning. Its proof is formed by concatenating the synthetic tactic steps applied by the generator. The resulting syntax tree—comprising a uniquely generated theorem name, theorem statement, and proof—is converted to raw syntax using Lean's pretty-printer.

Because our objective is to automatically extract training data from these files, we must ensure not only that each theorem is syntactically correct, but also that the file to which it is written is runnable without human intervention. Because the syntax and tactic behavior we use are sensitive to their execution environment (e.g., the behavior of `simp_arith` changes depending on the available simplification lemmas), we must ensure that Lean reconstructs our synthetic execution environment—which is derived in part from that of the library module from which our initial proof state is taken—when evaluating the output file. We accomplish this using Lean's metaprogramming framework to inspect the environment's imports, open modules, and namespaces, based on which we generate corresponding commands in a prepopulated header in each output Lean file. To avoid frequent recompilation, we reuse the same environment when generating theorems from each tactic state extracted from the same library module.

Even with these safeguards, however, some theorems may still fail to compile. This is because Lean has a rich and extensible notation system (Ullrich & de Moura, 2020), and the presence of user-defined or context-dependent notation further complicates the task of producing runnable code. As one of many examples, in-context hypotheses may contain type coercions whose target types are left implicit by the notation and cannot be inferred without the surrounding context. It is ultimately infeasible to account for every such possible notational complication that might arise, especially since modules may contain arbitrary user-defined notation. Accordingly, before the generator outputs a theorem, the candidate syntax string is evaluated as a Lean file would be: within the appropriate environment, the declaration is parsed and elaborated to ensure that the proposed theorem and proof are syntactically valid. Only after these checks succeed is the theorem written to the output file.

## 5 EXPERIMENTS

### 5.1 PERFORMANCE OF SYNTHETIC THEOREM GENERATION

We assessed the performance of our synthetic theorem generator by evaluating its throughput under various configurations of our deduplication and premise-selection procedures. For these throughput experiments, we selected input proof states from Mathlib due to its size and breadth. Our results are shown in Tables 1 and 2. These results do not reflect the maximum throughput of our generator and are instead intended to illustrate the relative effects of different configurations. Indeed, we are unable to compare the absolute throughput of our generator to that of other synthetic theorem generation approaches in Lean, such as those of Xin et al. (2024) and Ying et al. (2024), since those sources do not provide data regarding the time or computing resources required for generation. The full configuration of our experiments can be found in Appendix A.

As Table 1 demonstrates, the use of the ReProver model improves our generator's performance. Theorem output increases with the proportion of premises selected using LLM-based premise selection. We also found that the length of these theorems grows similarly, indicating that LLM selection

| % Random Premises | Synthetic Theorems | Average Proof Length |
|---|---|---|
| 0 | 1,008,188 | 6.63 |
| 20 | 931,707 | 6.63 |
| 50 | 751,442 | 6.70 |
| 80 | 365,675 | 6.37 |
| 100 | 122,639 | 6.00 |

Table 1: Throughput by varying the proportion of randomly- and LLM-selected premises.

| Premise Sample Size | Deduplication | Synthetic Theorems |
|---|---|---|
| 20 | No | 3,781,425 |
| 50 | Yes | 4,740,263 |
| 100 | Yes | 4,198,273 |

Table 2: Throughput by varying the number of premises sampled at each generation step.

facilitates longer chains of reasoning that may yield more complex theorems. These results are likely due to the fact that LLM-selected premises are relevant to in-context hypotheses, meaning that those hypotheses are more likely to satisfy the selected premises' antecedents to admit a new hypothesis. In contrast, given Mathlib's breadth, randomly selected premises are much less likely to be applicable to the hypotheses in a given proof state.

We found that the generator's throughput is especially sensitive to the number of premises it samples. Increasing the number of sampled premises increases the likelihood of a successful proof step but also the computation time necessary to identify one. As shown in Table 2, using an optimal number of sampled premises significantly increased the generator's throughput, even with the addition of deduplication to limit redundant outputs. This result, like that above, illustrates the critical role of successful premise applications in advancing synthetic proofs, even in the presence of other rewriting, simplification, and normalization tactics.

## 5.2 IMPACT OF SYNTHETICALLY GENERATED THEOREMS

To evaluate the impact of our synthetically generated theorems on LLMs' theorem-proving capabilities, we fine-tuned the Falcon2-11B model (Malartic et al., 2024) using synthetic theorems and measured its performance on the miniF2F benchmark (Zheng et al., 2022). For these experiments, we obtained input proof states to our generator from the Lean Workbook dataset (Ying et al., 2024) due to its focus on competition-style math. We applied several rounds of processing to the original Lean Workbook dataset to obtain our final input corpus, including deduplication, removal of invalid theorems, and decontamination with respect to the benchmark dataset. Details of this procedure are given in Appendix D.

Our experiment involved fine-tuning different baseline models with and without synthetically generated data. We established two baseline models: M1, the original Falcon2-11B model, and M2, the Falcon2-11B model fine-tuned on a mixed dataset containing theorems and proof artifacts (Han et al., 2022) extracted from Mathlib, and Lean textbooks. Details of this dataset are given in Appendix E. The Mathlib dataset contains 208 million tokens, while the mixed dataset contains 2.8 billion tokens. We then further fine-tuned the models on the Mathlib dataset either without synthetic data or with our synthetically generated dataset (approximately 1 billion tokens). The parameters used for generating the synthetic theorems and fine-tuning the models are detailed in Appendix A and Appendix C respectively. To generate training data from existing proofs in Mathlib and other sources, we traversed every tactic step in the corresponding Lean files and recorded the following data at each step:

- *GoalState*: the proof state prior to applying this tactic.
- *Tactic*: the syntax of the current tactic.
- *DeclUpToTactic*: the syntax, up to this tactic, of the declaration in which this proof occurs.

To evaluate the performance of models trained on synthetic datasets, we developed a parallel proof search infrastructure for generating these proofs. This infrastructure enables concurrent querying of LLMs for tactics and Lean for the next proof state. Using this setup, we query the LLMs to generate

| Model | Baseline Model | # of Theorems Proved | |
| --- | --- | --- | --- |
| | | w/o Synthetic Dataset | w/ Synthetic Dataset |
| M1 | Falcon2-11B | 91 / 244 (37.29%) | **94** / 244 (**38.52**%) |
| M2 | Falcon2-11B fine-tuned on mixed data | 93 / 244 (38.11%) | **96** / 244 (**39.34**%) |

Table 3: Number of theorems from miniF2F-test successfully proved by models fine-tuned on synthetic data generated from Lean Workbook. Examples of theorems proved only by models trained on the synthetic dataset are listed in Appendix G.

tactics and apply them to a proof state using REPL (Leanprover-Community, 2024b), a community-developed project capable of exporting and importing intermediate proof states. The order of tactics to search is determined by a best-first search strategy, where scores are based on the log probability of the LLM's generation. We host one LLM server per GPU and run one REPL process per CPU. In our experiments, we employed 40 CPU processes for proof state generation and 8 GPU processes for tactic generation. The proof search terminates upon generating a successful sequence of tactics that proves the original problem or upon reaching a timeout of 10 minutes.

We assessed the theorem-proving capabilities of our model variants by using them to generate proofs for 244 Lean theorems in the miniF2F-test dataset. The results of our experiments are shown in Table 3. These results demonstrate that the addition of synthetically generated datasets can significantly improve the performance of baseline models trained on different corpora. Specifically, for the model trained solely on the Mathlib dataset (M1), the synthetic data boosts theorem-solving performance from 91 to 94 theorems — an improvement that exceeds model performance by adding certain pre-existing datasets to train M2, which achieved 93 proved theorems. This indicates that our synthetic data not only matches but surpasses the value of existing datasets.

Additionally, when starting from a model trained on a mix of Mathlib and additional datasets (M2), fine-tuning with our synthetic data further improves performance, increasing the number of proved theorems from 93 to 96. This result highlights two key insights: (1) our synthetic data contains complementary information that enhances model performance even when existing data like Han et al. (2022) and textbook are already included, and (2) even for a model that has already achieved a strong baseline, incorporating our data leads to further gains, showing it can augment rather than replace or stagnate existing knowledge.

# 6 CONCLUSION

In this work, we have introduced a synthetic theorem generator capable of producing correct-by-construction theorems using forward reasoning in Lean. It achieves this through metaprogrammatic execution of forward-reasoning tactics, a search strategy optimized for diverse theorems that exclude irrelevant steps, and the integration of the ReProver LLM to select relevant premises.

We see several avenues to expand on the work we have presented. As noted in Section 4.3, our tactic execution procedure could be extended with additional forward-reasoning tactics, enabling the generation of proofs that use a greater variety of reasoning strategies. With a broader range of tactics from which to choose, the generator could additionally incorporate heuristics into tactic selection—beyond our existing premise selection—to more efficiently identify search paths likely to yield synthetic theorems of interest. Furthermore, as discussed in Section 4.4, a purpose-built premise-selection LLM might provide more effective suggestions of premises applicable to in-context hypotheses. Finally, because all theorems produced by our generator are correct by construction, our generator can be straightforwardly repurposed as a *conjecture*-generation tool simply by removing the proofs from the generator's output. These theorem statements could then be given to a deterministic or LLM-based proof-search procedure to prove using backward as well as forward reasoning, facilitating the creation of a dataset representative of a broader variety of Lean tactic proofs.

The availability of large quantities of high-quality training data is key to advancing LLMs' capabilities in formal theorem proving. To that end, with the goal of facilitating both the creation of new synthetic datasets and the furthering of this work, we release our synthetic theorem generator and other software including proof search code under the Apache 2.0 license.

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

# A  EXPERIMENTAL CONFIGURATION FOR SYNTHETIC THEOREM GENERATION

The experiments listed in Table 1 were conducted using the following parameters for the generator (any parameters not listed here or in that table were left at their default values):

| Parameter | Value |
|---|---|
| Depth | 10 |
| Minimum Depth | 3 |
| No Duplicate Theorems | True |
| No Duplicate Hypotheses | False |
| Per-Step Theorem Maximum | 100 |
| Premises | 100 |
| Premise Sample Size | 20 |

The experiments were run on Mathlib distributed across 20 machines, each with 72 vCPUs, with a 24-hour timeout.

The experiments listed in Table 2 were conducted using the following parameters:

| Parameter | Value |
|---|---|
| Depth | 10 |
| Minimum Depth | 3 |
| Per-Step Theorem Maximum | 500 |
| Premises | 100 |

"Deduplication" in Table 2 refers to the "No Duplicate Theorems" and "No Duplicate Hypotheses" options. The experiments were run on Mathlib distributed across 60 machines, each with 36 vCPUs, with a 24-hour timeout.

The experiments discussed in Section 5.2 were run across the Lean Workbook corpus, processed as described in Appendix D, on a single 192-vCPU machine and with the following parameters:

| Parameter | Value |
|---|---|
| Depth | 10 |
| Minimum Depth | 3 |
| Per-Step Theorem Maximum | 50 |
| Premises | 100 |
| Premise Sample Size | 50 |
| No Duplicate Theorems | True |
| No Duplicate Hypotheses | True |
| Premise Alternation | True |
| Minimization | Enabled, maximum minimized length = 3 |

# B  PROOF SYNTHESIS ALGORITHM PSEUDOCODE

In Algorithms 1 and 2, we detail the search algorithm used in proof synthesis. The actual Lean implementation is heavily monadic; we present here an imperative translation. We elide the full implementation of the function TACTICSFOR, which returns a stream of tactics potentially applicable at a provided proof state. Broadly, each entry in this stream consists of a tactic name together with in-context hypotheses—identified using simple heuristics—to which the tactic might be applicable. To improve performance, for candidate invocations of the `have` tactic, we record only the names of the candidate lemmas to attempt to apply; the exhaustive search for type-correct arguments to these lemmas only occurs when attempting to invoke the tactic.

**Algorithm 1** The generator's search procedure.

**Input:** $state_0$, an initial proof state drawn from Mathlib
1: $q \leftarrow$ empty min-heap of (proof state, tactic stream) pairs ordered by the number of tactics applied to the state so far
2: ENQ($q, (state_0, \text{TACTICSFOR}(state_0))$)
3: **while** $q$ is nonempty **and** more theorems are requested **do**
4:     $state_{\text{next}} \leftarrow$ DEQ($q$)
5:     $stack \leftarrow$ DFS($state_{\text{next}}$)
6:     **for** $(state, tactics)$ in $stack$ **do**
7:         ENQ($q, (state, tactics)$)
8:     **end for**
9: **end while**

---

**Algorithm 2** The DFS procedure invoked by the search routine above.

**Input:** a proof state $state$, a stream of tactics $tactics$, and a stack of (proof state, tactic stream) pairs $stack$
**Output:** a stack of (proof state, tactic stream) pairs representing unexplored tactic applications
1: $state' \leftarrow$ execute NEXT($tactics$) at $state$
2: **if** $state'$ was successfully produced by the tactic **then**
3:     PUSH($stack, (state, tactics)$)
4:     **if** $state'$ is at maximal depth **then**
5:         log the theorem generated at $state'$
6:     **else**
7:         **return** DFS($state', \text{TACTICSFOR}(state'), stack$)
8:     **end if**
9: **else**
10:     **if** $state'$ is deeper than the minimal depth **then**
11:         log the theorem generated at $state'$
12:     **else if** $stack$ is nonempty **then**
13:         $(state_{\text{backtrack}}, tactics_{\text{backtrack}}) \leftarrow$ POP($stack$)
14:         **return** DFS($state_{\text{backtrack}}, \text{backtrack}, stack$)
15:     **end if**
16: **end if**
17: **return** $stack$

---

## C  MODEL TRAINING SETUP

| | | M1 | | M2 | |
|---|---|---|---|---|---|
| Baseline | Epoch | N/A | | 10 | |
| Configuration | Tokens | | | 2.8B | |
| | | w/o Synthetic | w/ Synthetic | w/o Synthetic | w/ Synthetic |
| Fine-tuning | Epoch | 5 | 1 | N/A (Base model | 1 |
| Configuration | Tokens | 208M | 2B | is evaluated) | 2B |

Table 4: All training was conducted with a learning rate of $2 \times 10^{-5}$, a cosine learning rate scheduler, AdamW optimizer, and $0.05$ warmup ratio. The model is selected when the evaluation loss is minimized, with the corresponding epoch reported in the table.

We provide an overview of our training setup for both the baseline and fine-tuning configurations of our models. Table 4 summarizes the key parameters used in our experiments, including the number of epochs, token counts, and the presence or absence of synthetic data.

For our baseline configuration, we used two versions of the Falcon2-11B model: M1, which is the original Falcon2-11B model without any modifications, and M2, which is the Falcon2-11B model fine-tuned for 10 epochs on the 2.8B-token dataset described in Appendix E.

For M1, as shown in Table 4, we conducted fine-tuning experiments both with and without synthetic data. Without synthetic data, the model was trained for 5 epochs using 208M tokens (Mathlib). With synthetic data, we used a larger dataset of 2B tokens, consisting of a 1B-token synthetic dataset generated from the Lean Workbook dataset and a 1B-token Mathlib dataset. The Mathlib dataset was upsampled to achieve a 1:1 ratio between Mathlib-based tokens and Lean Workbook-based synthetic tokens. In this case, the minimum evaluation loss was reached after 1 epoch. It is important to note that for both configurations, we selected the model checkpoint that minimized the evaluation loss, which occurred at different epochs for each setup.

For M2, we evaluated the base model without additional fine-tuning in the scenario because the Mathlib dataset was already used in its training. When incorporating synthetic data, we fine-tuned M2 for 1 epoch using 2B tokens, mirroring the approach used for M1 with synthetic data.

It is important to note that all training sessions were conducted with consistent hyperparameters across models and configurations. We used a learning rate of $2 \times 10^{-5}$, a cosine learning rate scheduler, the AdamW optimizer, and a warmup ratio of $0.05$. The final model selection was based on the epoch that minimized the evaluation loss, as reported in the table.

## D  LEAN WORKBOOK DEDUPLICATION AND DECONTAMINATION

Before synthesizing theorems based on the proof states in the Lean Workbook dataset, we applied the following preprocessing steps: we upgraded the dataset to the Lean version targeted by our generator, 4.9.0; we removed duplicate theorems from the dataset; we removed theorems matching those in the miniF2F-test dataset; and we verified that theorems (and any proofs) successfully compiled. We detail each of these steps below.

**Lean version upgrade**: Lean Workbook targets Lean and Mathlib version 4.8.0-rc1, while our generator uses Lean and Mathlib 4.9.0. It was therefore necessary to update the Lean Workbook header file to account for recent changes to Mathlib.

**Deduplication**: To maximize the number of inputs to our generator, we used theorems from both the Lean Workbook and Lean Workbook Plus splits of the Lean Workbook corpus. However, since these splits were generated using overlapping natural-language theorem statements, it was necessary to apply deduplication to ensure that the same formal theorem did not appear multiple times. Moreover, we discovered that duplicate theorems also appear within the same split; we removed these as well.

We detected duplicates based on equality of the Lean `Expr` objects representing theorems' types. While a relatively rigid measure of similarity, it is more robust than direct syntactic comparison, as it detects variables that have been moved from parameter to argument position (i.e., before the colon to after the colon) or vice versa as well as alpha-equivalent theorems with differing variable names (since `Expr` values represent bound variables using de Bruijn indices).

**MiniF2F decontamination**: To avoid data contamination, we discarded all theorems in Lean Workbook that are duplicates of theorems in the miniF2F-test dataset. We used the same `Expr`-comparison technique as we did in our deduplication procedure: we initially recorded the types (as `Expr` values) of all theorems in miniF2F-test, then verified that the type of each theorem in Lean Workbook was not in this set before adding the theorem to our final dataset. During decontamination, we identified and removed 37 miniF2F-test theorems from the Lean Workbook dataset.

**Compilation verification**: We verified that each Lean Workbook theorem—including, if present, any proofs—successfully compiled prior to adding it to our dataset. Non-compiling theorems were discarded. Manual inspection revealed that a common cause of compilation errors was invalid syntax, usually due to the omission of leading portions of theorem statements. For instance, the following non-compiling theorem

```
theorem lean_workbook_40296 : ℝ) : (exp x + exp (-x)) / 2 ≤ exp (x^2 /
    2)  :=  by sorry
```

was likely intended to be written as follows:

```
theorem lean_workbook_40296 (x : ℝ) : (exp x + exp (-x)) / 2 ≤ exp (x^2
    / 2)  :=  by sorry
```

# E    DATA MIX

Our training dataset comprises a diverse mix of Lean 4-related content, totaling 2.768B tokens. This dataset includes the following components:

- Code: We incorporated Lean4 code from three primary repositories: Lean4, Mathlib4, Batteries.

- Proof Artifacts: This largest component of our dataset consists of nine distinct tasks designed to capture various aspects of theorem proving in Lean 4: a) next-lemma: predicting the next tactic given a proof state, b) premises: identifying global declarations needed to prove a given goal, c) local-premises: determining local premises required to prove the goal, d) local-lemmas: predicting local lemmas used to prove the goal, e) types: inferring the type of a given term based on the context, f) proof-terms: generating the entire proof term for a given goal, g) theorem-names: predicting the type of a given theorem name, h) docs: generating documentation strings for given declarations and types, and i) next-tactic-and-goal: predicting the next tactic to apply given a proof state.

- Textbooks: We included content from six key Lean 4-related textbooks: Functional Programming in Lean (Christiansen, 2023), Theorem Proving in Lean 4 (Avigad et al., 2024), Mathematics in Lean (Avigad & Massot, 2020), Glimpse of Lean (Massot, 2024), the Lean 4 Reference Manual (Leanprover-Community, 2024a), and Type Checking in Lean 4 (Bailey, 2024).

# F    SYNTHETIC THEOREM EXAMPLES

We provide below examples of synthetic theorems produced by our generator:

```
theorem choose_le_pow.step_2_ex_235 {α : Type u_1}
    [LinearOrderedSemifield α] (r : ℕ) (n : ℕ) :
    (n * r - n / r ^ n).choose ((r - 1) * Σ x ∈ Finset.range (1 + n), n
    / r ^ x) *
        ((n * r - n / r ^ n).choose ((r - 1) * Σ x ∈ Finset.range (1 +
    n), n / r ^ x) - 1).factorial =
      ((n * r - n / r ^ n).choose ((r - 1) * Σ x ∈ Finset.range (1 + n),
    n / r ^ x)).factorial :=
  by
  have aux_1 : (r - 1) * Σ i ∈ Finset.range n.succ, n / r ^ i ≤ n * r -
    n / r ^ n := Nat.pred_mul_geom_sum_le n r n
  have aux_2 : 0 < (n * r - n / r ^ n).choose ((r - 1) * Σ i ∈
    Finset.range n.succ, n / r ^ i) := Nat.choose_pos aux_1
  simp_arith at aux_2
  have aux_3 :
    ((n * r - n / r ^ n).choose ((r - 1) * Σ i ∈ Finset.range (n + 1), n
    / r ^ i)).choose 1 * factorial 1 *
        ((n * r - n / r ^ n).choose ((r - 1) * Σ i ∈ Finset.range (n +
    1), n / r ^ i) - 1).factorial =
      ((n * r - n / r ^ n).choose ((r - 1) * Σ i ∈ Finset.range (n + 1),
    n / r ^ i)).factorial :=
    Nat.choose_mul_factorial_mul_factorial aux_2
  simp_arith at aux_3
  ring_nf at aux_3
  exact aux_3
```

```
theorem step_173_ex_5 {ξ : ℝ} {u : ℤ} {v : ℤ} (hv : 2 ≤ v) (h :
    ContfracLegendre.Ass ξ u v) (hξ₀ : 0 < fract ξ) (u' : ℤ)
    (hu₀ : 0 < u') (huv : u' < v) (hu' : u' = u - ⌊ξ⌋ * v) :
    -u + u * v + (v * ⌊ξ⌋ - v ^ 2 * ⌊ξ⌋) < u * v - v ^ 2 * ⌊ξ⌋ :=
  by
  have aux_1 : v - 1 < v := sub_one_lt v
  have aux_2 : u' * (v - 1) < u' * v := mul_lt_mul_of_pos_left aux_1 hu₀
  rewrite [hu'] at aux_2
  ring_nf at aux_2
  ring_nf at aux_2
  exact aux_2

theorem mod_eq_zero.step_6_ex_8 {R : Type u} [EuclideanDomain R]
    {a : R} {b : R} (h : a % b = 0) :
    b ^ 2 | b * a :=
  by
  simp_arith at h
  have aux_1 : b * b | b * a := mul_dvd_mul_left b h
  ring_nf at aux_1
  exact aux_1
```

Below, we illustrate the intermediate proof states through which the generator progresses when producing the above theorem. (The proof code appears on the left, while the corresponding proof state appears on the right.) The generator begins with an initial proof state drawn from the proof of `EuclideanDomain.mod_eq_zero` in Mathlib. It then successively applies tactics until sufficiently many proof steps have been applied. Notice that the goal from the original proof state is retained but ignored, since the generator acts only on the hypotheses.

| | |
|---|---|
| ```theorem mod_eq_zero.step_6_ex_8 {R :     Type u} [EuclideanDomain R]     {a : R} {b : R} (h : a % b = 0) :     b | a :=   by``` | ```R : Type u_1 inst† : EuclideanDomain R a b : R h : a % b = 0 ⊢ b | a``` |
| ```theorem mod_eq_zero.step_6_ex_8 {R :     Type u} [EuclideanDomain R]     {a : R} {b : R} (h : a % b = 0) :     b | a :=   by   simp_arith at h``` | ```R : Type u_1 inst† : EuclideanDomain R a b : R h : b | a ⊢ b | a``` |
| ```theorem mod_eq_zero.step_6_ex_8 {R :     Type u} [EuclideanDomain R]     {a : R} {b : R} (h : a % b = 0) :     b | a :=   by   simp_arith at h   have aux_1 : b * b | b * a :=     mul_dvd_mul_left b h``` | ```R : Type u_1 inst† : EuclideanDomain R a b : R h : b | a aux_1 : b * b | b * a ⊢ b | a``` |
| ```theorem mod_eq_zero.step_6_ex_8 {R :     Type u} [EuclideanDomain R]     {a : R} {b : R} (h : a % b = 0) :     b | a :=   by   simp_arith at h   have aux_1 : b * b | b * a :=     mul_dvd_mul_left b h   ring_nf at aux_1``` | ```R : Type u_1 inst† : EuclideanDomain R a b : R h : b | a aux_1 : b ^ 2 | b * a ⊢ b | a``` |

Finally, once sufficiently many proof steps have been applied, the generator selects the type of the last-modified hypothesis to be the conclusion of the new theorem. The theorem statement is then updated accordingly, and the final `exact` tactic is added to the end of the proof:

```
theorem mod_eq_zero.step_6_ex_8 {R :
   Type u} [EuclideanDomain R]
   {a : R} {b : R} (h : a % b = 0) :
   b ^ 2 | b * a :=
  by
  simp_arith at h
  have aux_1 : b * b | b * a :=
    mul_dvd_mul_left b h
  ring_nf at aux_1
```

```
R : Type u_1
inst† : EuclideanDomain R
a b : R
h : b | a
aux_1 : b ^ 2 | b * a
⊢ b ^ 2 | b * a
```

```
theorem mod_eq_zero.step_6_ex_8 {R :
   Type u} [EuclideanDomain R]
   {a : R} {b : R} (h : a % b = 0) :
   b ^ 2 | b * a :=
  by
  simp_arith at h
  have aux_1 : b * b | b * a :=
    mul_dvd_mul_left b h
  ring_nf at aux_1
  exact aux_1
```

```
No goals
```

## G  THEOREMS PROVED AFTER TRAINING ON SYNTHETIC DATA

As indicated in Table 3, models fine-tuned using the synthetic dataset were able to prove more
theorems from the miniF2F-test benchmark than those fine-tuned only on data from Mathlib. Below,
we show examples of theorems from miniF2F-test that model M1 was able to prove after being fine-
tuned on the synthetic dataset, but which it failed to prove when fine-tuned only on Mathlib:

```
theorem mathd_algebra_263
  (y : ℝ)
  (h₀ : 0 ≤ 19 + 3 * y)
  (h₁ : Real.sqrt (19 + 3 * y) = 7) :
  y = 10 := by
rw [Real.sqrt_eq_iff_sq_eq h₀] at h₁
linarith
simpa using h₁
```

```
theorem amc12a_2002_p6
  (n : ℕ)
  (h₀ : 0 < n) :
  ∃ m, (m > n ∧ ∃ p, m * p ≤ m + p) := by
by_contra h
rw [not_exists] at h
simp only [not_and] at h
have h := h (n + 1) (by simp)
exact h ⟨1, by simp⟩
```

```
theorem mathd_algebra_113
  (x : ℝ) :
  x^2 - 14 * x + 3 ≥ 7^2 - 14 * 7 + 3 := by
aesop
by_cases h : x ≤ 7
nlinarith
case neg => exact le_of_lt (by nlinarith)
```

```
theorem induction_11div10tonmn1ton
   (n : ℕ) :
   11 | (10^n - (-1 : ℤ)^n) := by
cases' n with n
simp
nontriviality ℤ
induction' n with n ih
norm_num1
omega
```

## H  SYNTHETIC THEOREMS BY SUBJECT AREA

The following lists the number of theorems in our synthetic dataset synthesized from modules in each subject area represented in Mathlib. As these data show, the generator is capable of synthesizing theorems from a diverse range of proof states. While the exact content of a synthesized theorem will diverge somewhat from that of the initial proof state, these proof states influence the areas of math covered by the generated theorems, as the subject matter of the proof state will affect which lemmas are suggested by the premise-selection LLM, and the definitions referenced by the initial hypotheses are likely to appear in the final theorem statement.

| Subject | Theorems |
| --- | --- |
| Algebra | 636654 |
| AlgebraicGeometry | 11137 |
| AlgebraicTopology | 10358 |
| Analysis | 312373 |
| CategoryTheory | 15202 |
| Combinatorics | 97615 |
| Computability | 123594 |
| Control | 1228 |
| Data | 1663773 |
| Deprecated | 448 |
| Dynamics | 36626 |
| FieldTheory | 26104 |
| Geometry | 14623 |
| GroupTheory | 90086 |
| Init | 13893 |
| LinearAlgebra | 32062 |
| Logic | 39760 |
| MeasureTheory | 270751 |
| ModelTheory | 3474 |
| NumberTheory | 190169 |
| Order | 258210 |
| Probability | 39049 |
| RepresentationTheory | 198 |
| RingTheory | 192857 |
| SetTheory | 221814 |
| Tactic | 11492 |
| Testing | 1524 |
| Topology | 424135 |
| Util | 500 |

# I EXAMPLE OF A LEAN TACTIC PROOF

The following is an annotated example of a tactic proof in Lean. The `theorem` declaration gives the name and statement of the theorem to prove (i.e., `Nat.gcd_le_min` is a proof that $\gcd(m, n) \leq \min(m, n)$ for all positive natural $m$ and $n$). Each line after `by` invokes a tactic (e.g., `intro`, `rewrite`, `apply`, `exact`).

```
theorem Nat.gcd_le_min :
    ∀ m n : ℕ, m > 0 → n > 0 → gcd m n ≤ min m n := by
  -- Introduce the variables m and n and the named hypotheses hm : m > 0
     and hn : n > 0
  intro m n hm hn
  -- Rewrite the goal using the lemma le_min_iff : x ≤ min a b ↔ x ≤ a ∧
     x ≤ b
  rewrite [le_min_iff]
  -- Apply the introduction rule for conjunction
  apply And.intro
  -- Show the first conjunct (gcd m n ≤ m) using the lemma gcd_le_left
  · exact gcd_le_left n hm
  -- Show the second conjunct (gcd m n ≤ n) using the lemma gcd_le_right
  · exact gcd_le_right n hn
```

