# OpenReview forum: "Synthetic Theorem Generation in Lean"
_ICLR.cc/2025/Conference — Submitted to ICLR 2025_

### Official Review · Reviewer_KjJi · 2024-10-23

**Soundness:** 3
**Presentation:** 2
**Contribution:** 2
**Rating:** 5
**Confidence:** 5

**Summary:**

This paper presents an approach for generating synthetic formal theorems and their corresponding proofs by leveraging existing libraries through forward reasoning. Specifically, it focuses on modifying a given proof state by combining premise selection with a set of predefined tactics (e.g., `rewrite`, `simp_arith`). The method retrieves relevant lemmas and applies these tactics to generate new proof states. Additionally, the paper introduces several optimizations to enhance the diversity and quality of the generated theorems. These include a search algorithm to produce a broader variety of theorems within the same computational budget and the use of tactics like `omega` or `aesop` for proof minimization. By utilizing the Mathlib and Lean Workbook datasets, the approach can synthesize up to a million new theorems, expanding existing datasets. Experiments on miniF2F show that fine-tuning LLMs on this extended dataset leads to improved performance compared to fine-tuning on the original dataset alone.

**Strengths:**

> The paper is well-written and easy to follow.

> The proposed method is intuitive and is able to generate millions of new theorems from existing libraries. This addresses, to some extent, the critical challenge of data scarcity in formal theorem proving.

> The experiments show that fine-tuning on the extended dataset could lead to improved performance.

**Weaknesses:**

> Lack of Technical Novelty in Synthetic Theorem Generation

The approach of using forward reasoning for generating synthetic theorems and proofs is not particularly novel, as it has been widely explored in previous work. While this paper distinguishes itself by generating theorems from existing libraries rather than basic axioms in Lean, it still follows a similar, well-established pipeline.

> Potential Limited Diversity and Difficulty of Generated Theorems

The proposed method utilizes only five tactics applied in a linear fashion for synthetic generation, which may limit the variety of theorems it produces. As a result, the generated theorems tend to lack both diversity and complexity, particularly when compared to those written by humans. Additionally, the absence of examples of the generated theorems and their proofs in the paper makes it difficult to accurately evaluate the quality and difficulty of the generated datasets.

> Marginal Experimental Results

Despite the synthetic dataset being 10 times larger than the original (2B vs. 208M), the performance improvement is modest (e.g., the number of proved theorems only rises from 91 to 94, or 93 to 96). The paper also lacks an ablation study to evaluate the quality of the synthetic data or explain which theorems are newly proved. It’s unclear whether the additional proofs are due to the five tactics (or `omega`), or just the randomness of the experiments.

Nit: Figure 1, which explains basic background information, feels unnecessary. It would be more useful to include examples of the synthetic datasets or newly proved theorems in the miniF2F benchmark after fine-tuning on the extended datasets. Additionally, in Table 1, the label for the number of random premises seems to be fractional, but it’s listed as a percentage (%) in the row title, which could be misleading.

**Questions:**

> Could you conduct experiments or analysis to evaluate the quality of the synthetic datasets? For instance, how does the LLM perform when fine-tuned solely on the synthetic data? Additionally, could you compare the performance of a model fine-tuned on an equivalent-sized synthetic datasets to one trained on the real-world datasets?

> Could you also provide examples or newly proved theorems from the miniF2F-test, along with the specific tactics the model used to prove them? Additionally, do the theorems proved by the model fine-tuned on the extended datasets (e.g., 96 theorems) overlap with the theorems proved by its counterpart fine-tuned on the original datasets (e.g., 93 theorems), or are there distinct theorems unique to each?

---

> ### Author Response · Authors · 2024-11-25
>
> We thank the reviewer for their detailed feedback and suggestions. We respond below to the concerns and questions raised:
>
> > **W1.** Lack of Technical Novelty in Synthetic Theorem Generation
>
> We respectfully disagree with the conclusion that our work lacks novelty. We outline the novel aspects of our contribution below. Firstly, we are the first to realize such a system of forward-reasoning-based theorem synthesis in Lean's tactic mode, as also noted by reviewer tn5n. Since the majority of human-authored Lean proofs are written in tactic mode rather than as raw terms, this results in training data that is significantly more relevant to real-world proof tasks. Secondly, our approach is not restricted to a single mathematical domain, but is broadly applicable across the fields of math formalized in Mathlib, which represents a wide swath of modern mathematics. Lastly, we enhance our theorem synthesis procedure with LLM-based premise selection, which allows our tool to select relevant library lemmas across this broad array of mathematical disciplines, improving both efficiency and the relevance of the output theorems.
>
> > **W2.** Potential Limited Diversity and Difficulty of Generated Theorems
>
> While our current generator configuration uses five or seven tactics (depending on whether proof minimization is enabled), it is more extensible and capable of generating more diverse theorems than this may suggest. First, tactics can take arguments that significantly broaden the types of reasoning they can apply. The `have` tactic alone can apply more than 150,000 lemmas in the Mathlib library, which span much of modern mathematics. Secondly, our generator is extensible with additional forward-reasoning tactics—our architecture does not rely on the specific behavior of the five we selected. The number of forward-reasoning tactics available in Mathlib is currently limited, and we sought to choose a robust subset whose functionality did not overlap, but more forward-reasoning tactics can be easily added to the generator in cases where a premium is placed on the breadth of tactics used.
>
> We also believe that our synthetic theorems do exhibit diversity in several respects. Firstly, we promote diversity in the proofs we generate using the algorithm shown in Figure 3, which ensures that proofs diverge from one another as early as possible. Secondly, we generate theorems using initial proof states from across Mathlib, leading to synthetic theorems that address a wide array of subject areas in mathematics. We have added a table in Appendix H that lists the number of theorems in our synthetic dataset generated using initial states from each top-level Mathlib submodule (e.g., `Algebra`, `Analysis`, `Topology`), illustrating this subject-matter diversity. The theorems can also exhibit user-specified degrees of complexity, since the number of reasoning steps they employ is configurable in the generator and can be set to arbitrarily large values. With regard to your final concern in this item, we have also added examples of synthetically generated theorems in Appendix F.

---

> ### Author Response · Authors · 2024-11-25
>
> > **W3.** Marginal Experimental Results
>
> While the performance improvement from our synthetic data was modest, we believe its consistency provides support for the utility of our synthetic theorems. We note that there are avenues to diversify and enhance the generated data—such as expanding the number of forward-reasoning tactics as they become available in Mathlib, or replacing ReProver with a purpose-built premise-selection model (we selected ReProver for its ease of use and ubiquity in this area, though it is not directly optimized for this task)—that may improve the utility of the generated theorems without modifying the overall architecture of the generator. It is this broader architectural contribution that we believe distinguishes our work; as we noted in our response to item W1 above, there are multiple novel aspects we present in this regard, including our search/synthesis strategy (§§4.2–3), tactic-execution/simulation scheme (§4.1), LLM-based lemma selection (§4.4), and verification pipeline (§4.5).
>
> With respect to your concern regarding ablation studies, we did evaluate the model both with and without synthetic training data, as shown in Table 3. We did not include the performance of the base model without any fine-tuning because its performance was quite poor. We also did not evaluate the model when trained solely on synthetic data because the intent of our dataset is to serve as a complement to existing real-world data, providing additional examples of key reasoning strategies in Lean, but not as an all-encompassing corpus of Lean/Mathlib reasoning tactics. Since our synthetic theorems are targeted in this manner, they are not designed to be the sole source of training data for a model.
>
> We appreciate your comment that your evaluation of our data would be aided by assessing the newly proved theorems. To this end, we have added in Appendix G examples of theorems proved with but not without synthetic training data: as these demonstrate, the newly proved theorems are from multiple distinct domains and use a variety of tactics. Moreover, the performance of models fine-tuned with synthetic data consistently exceeded that of models fine-tuned only on Mathlib, supporting the efficacy of the synthetic data.
>
> > **W4.** Nit: Figure 1, which explains basic background information, feels unnecessary. It would be more useful to include examples of the synthetic datasets or newly proved theorems… Additionally, in Table 1, the label for the number of random premises seems to be fractional, but it’s listed as a percentage (%) in the row title, which could be misleading.
>
> We appreciate your suggestion regarding the addition of examples of synthetic theorems and newly proved theorems, and we have added these to Appendices F and G. We have added a figure with a concrete example of a synthetically generated theorem, per your suggestion, and have moved the former Figure 1 to an appendix. Thank you also for catching the discrepancy in Table 1; we have corrected this error.
>
> > **Q1.** Could you conduct experiments or analysis to evaluate the quality of the synthetic datasets? For instance, how does the LLM perform when fine-tuned solely on the synthetic data? Additionally, could you compare the performance of a model fine-tuned on an equivalent-sized synthetic datasets to one trained on the real-world datasets?
>
> The intent of our dataset is to serve as a complement to existing real-world data, providing additional examples of key reasoning strategies in Lean. However, since our synthetic theorems are targeted in this manner, they are not intended to provide an exhaustive treatment of all possible tactics and reasoning strategies that are possible with Lean/Mathlib, and therefore are not designed to be the sole source of training data for a model.
>
> > **Q2.** Could you also provide examples or newly proved theorems from the miniF2F-test, along with the specific tactics the model used to prove them? Additionally, do the theorems proved by the model fine-tuned on the extended datasets…overlap with the theorems proved by its counterpart…?
>
> We have added in Appendix G several examples of theorems that were proved by a model fine-tuned on the synthetic datasets but not proved by the same model fine-tuned only on Mathlib. To specifically address your second question, there were also theorems proved by the model fine-tuned only on Mathlib data that were not proved by the model fine-tuned on synthetic data—neither set of proved theorems (with synthetic fine-tuning data or with Mathlib-only fine-tuning data) was a subset of the other.

---

> > ### Comment · Reviewer_KjJi · 2024-11-26
> >
> > Thank you for your response; it partially addresses some of my concerns, so I have raised my score to 5. While I understand that the proposed datasets are intended to complement existing manually constructed datasets, I find that they lead to only very modest improvements in the LLMs' performance. This raises questions about the overall impact and significance of the datasets in improving the theorem-proving capabilities of LLMs.
> >
> > Moreover, although some theorems could be proved only after fine-tuning on the extended dataset, others fail to be proved after fine-tuning. Could you please explain why some theorems fail to be proved only when trained on the extended dataset? Is this due to possibly randomness in the experiments?
> >
> > Additionally, it seems that the authors rely solely on timeout as the evaluation metric. Could you also report Pass@k metrics or indicate how many search attempts the proposed methods can perform within a 10-minute time limit? This would offer a clearer understanding of the method's efficiency and practical utility.

---

> > > ### Author Response · Authors · 2024-11-28
> > >
> > > We appreciate these thoughtful comments regarding the evaluation of our synthetic datasets. While the improvements in proof search performance may appear modest, we believe they are meaningful given that they were achieved through synthetic data alone, without any additional techniques. The consistent improvements across different evaluations suggest that our synthetic data generation approach provides value as a complementary method to existing datasets.
> > >
> > > Regarding the cases where theorems become unprovable after fine-tuning on the extended dataset, we have investigated several potential causes. While experimental randomness may play a role, we believe a key factor is the lack of model alignment after fine-tuning. Our synthetic data focuses on improving the model's understanding of Lean and mathematical reasoning fundamentals, but was not specifically aligned for competitive mathematical problem-solving. This suggests an opportunity to combine our synthetic data approach with model alignment to potentially achieve better results.
> > >
> > > Regarding evaluation metrics, our approach differs slightly from the traditional Pass@k metric since our model generates tactics sequentially rather than the entire proof. During the 10-minute timeout period, our models generate an average of 43873 tactics at timeout for fine tuned M1 on synthetic dataset, and an average of 61386 tactics at timeout for fine tuned M2 on synthetic dataset. We believe these metrics offer a more complete picture of our method's practical performance.

---

### Official Review · Reviewer_incY · 2024-11-02

**Soundness:** 3
**Presentation:** 3
**Contribution:** 3
**Rating:** 5
**Confidence:** 4

**Summary:**

This paper presents a synthetic generator to create diverse new theorem and proof data for Lean 4 by forward reasoning and premise selection in existing libraries like Mathlib. By applying the generator to Lean Mathlib and Lean Workbook dataset, the authors demonstrate improvements in theorem-proving performance, with a fine-tuned Falcon2-11B model showing increased accuracy from 38.1% to 39.3% on the miniF2F benchmark.

**Strengths:**

1. The paper is logically structured and comprehensively explains the generator’s architecture, from premise selection to proof synthesis search and minimization. The methodology for generating synthetic theorems through forward reasoning is well-justified.
2. The approach’s reliance on premise selection using the LeanDojo ReProver model ensures theorems remain relevant and mathematically grounded, leading to meaningful performance gains on the miniF2F benchmark.
3. This paper significantly advances synthetic data generation for theorem proving by moving beyond simple mutation to generate genuinely new theorems and mitigate the data scarcity issue in formal theorem proving. The use of forward reasoning allows for the production of diverse, novel theorems, which may expand formal datasets in ways that mutation alone may not achieve.

**Weaknesses:**

1. Marginal Improvement in Benchmark Performance: Despite generating millions of synthetic theorems and proofs, the fine-tuning of these theorems leads to only a modest improvement in miniF2F performance for the Falcon2-11B model trained on a mixed dataset (from 38.1% to 39.3%). This improvement is notably lower than state-of-the-art approaches such as DeepSeekProver v1.5 and InternLM Prover v2.5, which achieve over 60% on miniF2F dataset. This raises concerns regarding the quality and utility of the generated theorems for practical theorem proving.
2. Lack of Quality Metrics for Synthetic Theorems: The paper does not provide specific metrics or validation methods to assess the quality or relevance of the generated synthetic theorems. The absence of such metrics makes it difficult to determine whether the synthetic data aligns well with mathematically significant problems or contains inherent limitations affecting model performance.
3. Computational Requirements: The generation process requires significant computational resources (60 * 36 vCPU for 24 hours), even if the search depth is not very high (10). It may potentially limit accessibility for broader use.

**Questions:**

1. Given the modest improvement in miniF2F accuracy, are there any metrics or quality checks in place to evaluate the significance and validity of the generated theorems and proofs beyond their correctness by construction?
2. Which specific theorems in miniF2F were newly proved by the Falcon2-11B model fine-tuned with synthetic data? These examples could help clarify the types of problems that benefit from synthetic training.
3. Could this approach be modified or enhanced to better align with the needs of SOTA theorem-proving systems, potentially addressing the disparity in miniF2F performance compared to SOTA models? Is there any specific reason to use Falcon2-11B compared to other models?
4. In Appendix B, you assumed a TacticsFor function in Algorithm A that returns a set of applicable theorems at a proof state. How is this function implemented? Is it time-consuming since it may require to iterate over all potential theorems and variable assignments?
[1] Xin, Huajian, et al. "DeepSeek-Prover-V1.5: Harnessing Proof Assistant Feedback for Reinforcement Learning and Monte-Carlo Tree Search." arXiv preprint arXiv:2408.08152 (2024).
[2] Ying, Huaiyuan, et al. "Lean Workbook: A large-scale Lean problem set formalized from natural language math problems." arXiv preprint arXiv:2406.03847 (2024).

---

> ### Author Response · Authors · 2024-11-25
>
> We thank the reviewer for their in-depth feedback and suggestions. We respond below to the concerns and questions raised:
>
> > **W1.** …the fine-tuning of these theorems leads to only a modest improvement in miniF2F performance…notably lower than state-of-the-art approaches such as DeepSeekProver v1.5 and InternLM Prover v2.5, which achieve over 60% on miniF2F dataset.
>
> While our particular experiments did not achieve the same performance as SOTA approaches like DeepSeekProver and InternLM, there are several factors that contribute to this disparity; because of this, and because of the more broadly applicable nature of our work, we believe our contributions are still significant despite these differences in absolute performance.
>
> * As we elaborate upon in our response to item Q3 below, our focus was not on the architecture of our final proof-search model but rather on the synthetic theorem generator itself. Accordingly, we opted to build and fine-tune a proof-search approach representative of common prior work; our architecture was a parallelized version of that used by LeanDojo. Thus, it was expected that our performance would not match that of SOTA architectures like those used by DeepSeekProver and InternLM; instead, we aimed to show that synthetic data could be successfully employed to improve theorem-proving ability, which we believe the data in §5.2 evince.
>
> * As we note in §6, the diversity of our generator’s output could be further enhanced with expanded forward-reasoning capabilities in Mathlib. Our generator is capable of supporting a wide range of forward-reasoning strategies; currently, though, there is a relative paucity of strictly forward-reasoning tactics in Mathlib. This is not a restriction of our generator *per se*, but rather reflects the fact that many tactics in Mathlib operate exclusively or partially backward. However, many backward-reasoning simplification and rewriting strategies can be adapted to forward reasoning by performing the same operations at a hypothesis instead of a goal (this is how we were able to employ typically backward-reasoning tactics like `rewrite` and `simp` in a forward-reasoning manner). Therefore, if and as more forward-reasoning functionality is added to Mathlib, our generator could produce more varied theorems, equipping LLMs to reason using a wider range of tactics, while using the same architecture we have already presented.
>
> * As we note in §§4.4 and 6, we relied on the LeanDojo ReProver model—which is limited compared to SOTA—for premise selection. We selected this model for its ease of use and the fact that it was trained on a related—but not precisely aligned—task. A more powerful or purpose-designed model could suggest more relevant theorems, producing proofs that more robustly demonstrate the effective use of both library lemmas and built-in tactics. Nonetheless, we believe the positive results we saw even using the ReProver model provides evidence for the utility of our approach irrespective of the additional benefit conferred by the selection of premise-selection model.
>
> Moreover, our primary contribution is our novel architecture for generating synthetic training data through forward reasoning in Lean, including a diversity-optimized search strategy (§§4.2–3), lightweight tactic-execution/simulation scheme (§4.1), LLM-based lemma selection (§4.4), and verification pipeline that accounts for Lean’s significant notational extensibility (§4.5). We believe these contributions, whose efficacy is supported by a modest but consistent performance improvement, still offer substantive benefits for LLM-based theorem proving.

---

> ### Author Response · Authors · 2024-11-25
>
> > **W2.**  The paper does not provide specific metrics or validation methods to assess the quality or relevance of the generated synthetic theorems.
>
> We appreciate the reviewer's concern about quality assessment of the synthetic theorems. While we acknowledge that developing comprehensive quality metrics for theorem generation remains an open research challenge, we provide several concrete measures to assess our approach:
>
> First, the coverage table presented in Appendix H serves as a quality indicator. This table demonstrates how our synthetic theorems span different mathematical fields, providing a quantitative measure of the diversity and relevance of the generated content.
>
> Second, we compare the complexity of proofs that M1 can generate with and without synthetic data fine-tuning. Our results show that models trained with synthetic data are capable of producing more complex proofs. This serves as a proxy measure for the quality and utility of the synthetic training data.
>
> Third, one metric we already recorded in our initial experiments—as displayed in §5.1—is the mean length of the output proofs. Since each line corresponds to an additional reasoning step required to deduce the conclusion from the assumptions, this metric serves as a proxy for the complexity/difficulty of the proof. Because the desired range of generated proof lengths is configurable, the generator is capable of creating proofs of effectively arbitrary complexity with respect to this metric.
>
> We agree that developing more sophisticated quality metrics for synthetic theorems is an important direction for future research. However, this represents a broader challenge in the field of automated theorem proving that extends beyond the scope of our current work. Our focus in this paper is on demonstrating the practical utility of our synthetic approach through measurable improvements in model performance.
>
> > **W3.** The generation process requires significant computational resources
>
> While our experimental setup did require significant computational resources, these can be reduced by modifying the generator’s configuration, such as using fewer premises or disabling re-elaboration during the verification phase. Additionally, the process of proof search frequently yields diminishing returns: initially, easy-to-synthesize proofs are rapidly produced; later, as the generator is left to find more challenging proofs (e.g., ones where premise selection yields many “near-miss” lemmas or in Mathlib modules with large numbers of proof steps that yield few proofs), the rate of proof production slows considerably. We observed this repeatedly in our experiments. Therefore, we anticipate that one could generate a significant fraction of the proofs we produced without extensive computational resources. Moreover, our experimental setup was batched, such that we could not granularly determine whether all vCPUs remained in use for the entire 24 hours; it is possible that certain machines with long-running tasks continued while those that had completed theorem generation sat idle, which would yield an inflated account of our resource utilization.
>
> > **Q1.** …are there any metrics or quality checks in place to evaluate the significance and validity of the generated theorems and proofs beyond their correctness by construction?
>
> Please refer to our response to item W2 above.
>
> > **Q2.** Which specific theorems in miniF2F were newly proved by the Falcon2-11B model fine-tuned with synthetic data? These examples could help clarify the types of problems that benefit from synthetic training.
>
> Thank you for this suggestion. We have provided below a few examples of newly proved problems from minif2f. We have also added Appendix G, which includes full newly-proven theorems and their proofs from a model fine-tuned on synthetic data.
>
> ```lean
> theorem mathd_algebra_263
>   (y : ℝ)
>   (h₀ : 0 ≤ 19 + 3 * y)
>   (h₁ : Real.sqrt (19 + 3 * y) = 7) :
>   y = 10
>
> theorem amc12a_2002_p6
>   (n : ℕ)
>   (h₀ : 0 < n) :
>   ∃ m, (m > n ∧ ∃ p, m * p ≤ m + p)
>
> theorem mathd_algebra_113
>   (x : ℝ) :
>   x^2 - 14 * x + 3 ≥ 7^2 - 14 * 7 + 3
>
> theorem induction_11div10tonmn1ton
>   (n : ℕ) :
>   11 ∣ (10^n - (-1 : ℤ)^n)
> ```

---

> ### Author Response · Authors · 2024-11-25
>
> > **Q3.** Could this approach be modified or enhanced to better align with the needs of SOTA theorem-proving systems, potentially addressing the disparity in miniF2F performance compared to SOTA models? Is there any specific reason to use Falcon2-11B compared to other models?
>
> We believe that our techniques are applicable to a wide range of models that require formal theorem statements and proofs in Lean for training. There is nothing specific to our synthetic generation approach that requires our specific model architecture.
>
> Our intent was not to focus on the architecture of the proof-search model itself, but rather on the relative impact of these synthetic data compared to training with organic data alone. Therefore, for our experiments, we opted to use a proof-search architecture representative of many commonly employed in prior work, even if this does not match the precise architectures of the latest SOTA systems. We use the same proof search algorithm as LeanDojo [1], with the only modification being that we implemented a parallel version of it. Since synthetic data did improve theorem-proving performance in this setting, we believe our synthetic generator would also be beneficial in SOTA theorem-proving systems, since, as noted, no aspect of the synthetic data is tied to our particular model configuration.
>
> We opted to use the Falcon2-11B model mainly because of our familiarity with the model and the fact that it is one of several open-sourced SOTA pre-trained LLMs. However, our methods are not specific to the Falcon model, and these same techniques could be leveraged to fine-tune other LLMs as well.
>
> > **Q4.** In Appendix B, you assumed a TacticsFor function in Algorithm A that returns a set of applicable theorems at a proof state. How is this function implemented?
>
> The `TacticsFor` function is straightforward and not computationally intensive, which is why we elected not to give explicit pseudocode; however, we have now added additional details to Appendix B to clarify this function’s behavior. To address your specific concern, it does not iterate over all possible arguments to the selected library lemmas. Instead, in the stream of tactics to apply, we represent `have` tactics simply by the library lemmas that the generator will attempt to apply; the generator only attempts to fill in the arguments to those lemmas when "executing" the tactic (i.e., line 1 of Algorithm 2)—this is the computationally expensive step in the procedure.
>
> [1] Kaiyu Yang, Aidan Swope, Alex Gu, Rahul Chalamala, Peiyang Song, Shixing Yu, Saad
> Godil, Ryan J Prenger, and Animashree Anandkumar. Leandojo: Theorem prov-
> ing with retrieval-augmented language models. In A. Oh, T. Naumann, A. Glober-
> son, K. Saenko, M. Hardt, and S. Levine (eds.), Advances in Neural Information Pro-
> cessing Systems, volume 36, pp. 21573–21612. Curran Associates, Inc., 2023.

---

> > ### Comment · Reviewer_incY · 2024-11-27
> >
> > Thank you for your detailed response and the added section in the appendix, which addressed most of my questions and concerns. While there is a lot of room for further improvements on the qualities/effects of synthetically generated theorems and performance on minif2f, this work still presents a meaningful step toward addressing data scarcity in formal theorem proving. I am confident that with the suggested refinements, this work can significantly impact the field. Thus I raise my rating from 3 to 5.

---

> > > ### Author Response · Authors · 2024-11-28
> > >
> > > Thank you for your feedback and for taking the time to review our responses. We are glad to hear that our responses addressed many of your questions and concerns, especially those regarding our work's impact in the field. We appreciate your suggestion to further refine our quality and performance analysis and will be sure to incorporate this into the final version of the paper.
> > >
> > > Thank you once again for your reviews and for increasing your rating.

---

### Official Review · Reviewer_tn5n · 2024-11-03

**Soundness:** 3
**Presentation:** 2
**Contribution:** 2
**Rating:** 5
**Confidence:** 4

**Summary:**

This paper introduces a novel approach to synthesizing new theorems in Lean. It utilizes forward reasoning tactics to construct new theorems based on a collection of existing proof states from the Mathlib library or Lean workbook. Additionally, it incorporates premise selection and proposes innovative search strategies to ensure the diversity and usefulness of the generated theorems. By using these newly generated theorems as complementary training corpus, the resulting theorem proving agent demonstrates an improvement of approximately 1% (3 problems) over the ablated setting.

**Strengths:**

- Given the current status of formal theorem proving, the need for a method capable of synthesizing theorems across a broad domain in Lean is critical. Therefore, the motivation for this work is compelling, and it represents a significant attempt to address the theorem generation problem in a wide context.

- This paper explores a novel approach to constructing new theorems in Lean. The use of forward reasoning to construct theorems has previously only been applicable in term mode, and I believe this is the first work to introduce this approach in tactic mode.

- The paper introduces several interesting components to ensure that the generated theorems are diverse and useful while eliminating unnecessary steps in the generated proofs.

**Weaknesses:**

- The usefulness of these synthesized theorems is a significant concern. The ablation performance using these generated theorems is quite weak, with only 3 additional theorems proven in miniF2F in both settings (M1 and M2). This is notable given that the tokens for the synthesized proofs amount to approximately 1 billion, compared to 208 million tokens in Mathlib.

- (Please correct me if I'm wrong) It appears that the tactics used to develop these forward reasoning steps are limited to a small subset, with only 5 tactics described in lines 296 to 301. Does this imply that the generated theorems will also only employ these 5 tactics? Considering the large variety of backward reasoning tactics available in Lean, generating theorems based solely on this limited subset of forward reasoning tactics could result in a biased set of synthetic data. I assume this is why the performance with these synthetic data is not more prominent.

- The paper is not very well written. Although the logic is easy to follow, the graphs are not particularly illustrative and lack detail. It would be beneficial to include a concrete, illustrative example when describing the method, as this would make the paper much easier to understand.

**Questions:**

- Is it true that the base tactics used for theorem generation (excluding variants with specific parameters) consist of only five (or a slightly larger number)? Will the final generated theorems be limited to tactics from this set?

- Could you provide some concrete examples of the generated theorems? A detailed process showing how one of these theorems is created would be particularly helpful.

- In lines 73–76, references for "Some" and "Other techniques" should be added.

- In lines 115–116, I believe the references "(An et al., 2024; Polu & Sutskever, 2020; Zombori et al., 2021)" are misplaced. These do not pertain to methods used for generating theorem statements.

---

> ### Author Response · Authors · 2024-11-25
>
> We thank the reviewer for their thorough feedback and suggestions. We respond below to the concerns and questions raised:
>
> > **W1.** The usefulness of these synthesized theorems is a significant concern.
>
> We acknowledge that the absolute increase in the number of theorems proved is limited in our particular experimental configuration. However, our work more broadly presents an architecture for forward reasoning-based tactic-proof synthesis in Lean using a novel search strategy (§§4.2–3), lightweight tactic-execution/simulation scheme (§4.1), LLM-based lemma selection (§4.4), and verification pipeline that accounts for Lean’s significant notational extensibility (§4.5). We believe that these broader architectural contributions to theorem synthesis, in conjunction with our experimental findings that show a modest but nonetheless consistent improvement in performance, still provide substantive benefits for synthetic data generation. Indeed, there are several factors unrelated or nonessential to our general architecture that likely impacted our experimental results:
>
> * As we elaborate upon in our response to item W2 below, the number of forward-reasoning tactics in Lean/Mathlib is currently limited, which does prevent us from generating certain types of proofs (e.g., structured induction proofs) that might provide useful training examples. While, as we note in our response to W2, our existing tactics do still offer diverse reasoning strategies. The various potential extensions of our tactic inventory we discuss in that response could increase the utility of the synthesized theorems—using the same architecture we have presented here.
>
> * As we note in §§4.4 and 6, improving the model used for premise selection could lead to more relevant theorem selection, producing proofs that demonstrate the effective use of both library lemmas and built-in tactics. While we rely on ReProver because it is trained on a related task and works easily out-of-the-box, it is not—as we note in §4.4—specifically trained for this task and, as a result, may produce less relevant lemma suggestions than a purpose-built LLM. Even so, the positive results we saw even with the ReProver model suggests the efficacy of our general approach, even without the additional benefit of a more powerful premise-selection model.
>
> * For our experiments, we opted to use a proof-search architecture representative of techniques commonly employed in prior work rather than the latest SOTA approaches, since our focus was on synthetic theorem generation rather than proof search. (Our approach was a modified version of that used by LeanDojo ReProver [10].) Accordingly, its performance was constrained relative to SOTA works.

---

> > ### Author Response · Authors · 2024-11-25
> >
> > > **W2.** Does this imply that the generated theorems will also only employ these 5 tactics?
> >
> > The generator is not architecturally limited to only employing the five tactics mentioned, as the system is more flexible and extensible than it might initially appear. Currently, our generator employs five forward-reasoning tactics (`have`, `rewrite`, `simp_arith`, `norm_num1`, and `ring_nf`) in addition to two optional proof-minimization tactics (`omega` and `aesop`), the behavior of each of which we describe in §4.3. However, there are two key ways in which our generator is more general than these figures might suggest:
> >
> > * Tactics can take arguments that significantly broaden the types of reasoning they can apply. For instance, although `have` is technically a single tactic, it is capable of applying any of the more than 150,000 lemmas in the Mathlib library, allowing it to invoke a wide range of inference strategies. As an example of the wide range of reasoning strategies a single tactic can support, using the `have` tactic with the lemma `Nat.lcm_pos` allows Lean to infer that `lcm m n` is positive from the facts that `m` and `n` are (a basic number-theoretic inference), while using the same `have` tactic with the lemma `Monotone.ae_differentiableAt` allows Lean to infer that a function `f` is differentiable Lebesgue-almost everywhere from the fact that `f` is monotone (a nontrivial result from analysis).
> >
> > * Our generator's infrastructure does not crucially depend upon the behavior of these particular tactics and could easily be extended with additional forward-reasoning tactics. There are currently a limited number of forward-reasoning tactics available in Mathlib, of which we aimed to select a robust subset with minimal overlap in functionality. With Mathlib's current inventory of tactics, we determined that the computational cost of adding any of the remaining forward-reasoning-capable tactics (e.g., `norm_cast0`) was not justified by the limited number of substantively distinct reasoning strategies such tactics would afford, especially given our desire to generate a large synthetic corpus and the diversity afforded by the tactics already employed. For use cases that place a premium on tactic diversity, however, extensibility with additional tactics is possible, especially if more forward-reasoning tactics or forward-reasoning modes for existing tactics are added to Mathlib.
> >
> > > **W3.** …the graphs are not particularly illustrative and lack detail. It would be beneficial to include a concrete, illustrative example when describing the method, as this would make the paper much easier to understand.
> >
> > We have added a figure that includes a simple concrete example of the method of theorem synthesis. We have also added concrete examples of synthetically generated theorems to Appendix F, where we also give an account of the process by which they were generated to supplement the graphical illustrations.
> >
> > > **Q1.** Is it true that the base tactics used for theorem generation (excluding variants with specific parameters) consist of only five (or a slightly larger number)? Will the final generated theorems be limited to tactics from this set?
> >
> > Please refer to our response to item W2 above.
> >
> > > **Q2.** Could you provide some concrete examples of the generated theorems? A detailed process showing how one of these theorems is created would be particularly helpful.
> >
> > Please refer to our response to W3 and the appendix we reference there.
> >
> > > **Q3.** In lines 73–76, references for "Some" and "Other techniques" should be added.
> >
> > [1,2,3,4] are works that generate proofs by performing proof search from conjectures from random sampling or LLM-based methods. [5,6,7,8,9] are works that generate new theorems and their proofs by applying inference rules to reason forward from existing theorems. We have added these references in the updated paper.

---

> ### Author Response · Authors · 2024-11-25
>
> > **Q4.** In lines 115–116, I believe the references "(An et al., 2024; Polu & Sutskever, 2020; Zombori et al., 2021)" are misplaced. These do not pertain to methods used for generating theorem statements.
>
> We appreciate this detailed feedback. Indeed, the citation of Polu & Sutskever here was misplaced, as we only intended to cite this work in the subsequent paragraph; in their §4.6, those authors discuss procedures for generating random theorem statements in the domains of $n$-digit arithmetic and ring algebra using a forward-reasoning-based approach. We believe that the citations of An et al. and Zombori et al. in their current location are relevant, as each paper discusses (though not as its primary contribution) a method of generating known-true theorem statements within a fixed domain:
>
> * An et al., 2024: §3 describes a procedure for generating theorem statements in intuitionistic propositional logic.
> * Zombori et al., 2021: §4 alludes to a procedure by which arithmetic problems (i.e., theorem statements) are randomly generated; the generator itself can be found in the [project repository](https://github.com/atpcurr/atpcurr).
>
> [1] Chenyang An, Zhibo Chen, Qihao Ye, Emily First, Letian Peng, Jiayun Zhang, Zihan Wang, Sorin Lerner, and Jingbo Shang. Learn from failure: Fine-tuning llms with trial-and-error data for intuitionistic propositional logic proving. arXiv preprint arXiv:2404.07382, 2024.
> [2] Huajian Xin, Daya Guo, Zhihong Shao, Zhizhou Ren, Qihao Zhu, Bo Liu, Chong Ruan, Wenda Li, and Xiaodan Liang. Deepseek-prover: Advancing theorem proving in llms through large-scale synthetic data. arXiv preprint arXiv:2405.14333, 2024.
> [3] Huaiyuan Ying, Zijian Wu, Yihan Geng, Jiayu Wang, Dahua Lin, and Kai Chen. Lean workbook: A large-scale lean problem set formalized from natural language math problems, 2024.
> [4] Zsolt Zombori, Adrián Csiszárik, Henryk Michalewski, Cezary Kaliszyk, and Josef Urban. Towards finding longer proofs. In Anupam Das and Sara Negri (eds.), Automated Reasoning with Analytic Tableaux and Related Methods, pp. 167–186, Cham, 2021.
> [5] Vlad Firoiu, Eser Aygun, Ankit Anand, Zafarali Ahmed, Xavier Glorot, Laurent Orseau, Lei Zhang, Doina Precup, and Shibl Mourad. Training a first-order theorem prover from synthetic data. In International Conference on Learning Representations Workshop on Mathematical Reasoning in General Artificial Intelligence, 2021.
> [6] Guillaume Lample, Timothee Lacroix, Marie anne Lachaux, Aurelien Rodriguez, Amaury Hayat, Thibaut Lavril, Gabriel Ebner, and Xavier Martinet. Hypertree proof search for neural theorem proving. In Alice H. Oh, Alekh Agarwal, Danielle Belgrave, and Kyunghyun Cho (eds.), Advances in Neural Information Processing Systems, 2022.
> [7] Stanislas Polu and Ilya Sutskever. Generative language modeling for automated theorem proving. arXiv preprint arXiv:2009.03393, 2020
> [8] Trieu H. Trinh, Yuhuai Wu, Quoc V. Le, He He, and Thang Luong. Solving olympiad geometry without human demonstrations. Nature, 625(7995):476–482, 2024.
> [9] Mingzhe Wang and Jia Deng. Learning to prove theorems by learning to generate theorems. In H. Larochelle, M. Ranzato, R. Hadsell, M.F. Balcan, and H. Lin (eds.), Advances in Neural Information Processing Systems, volume 33, pp. 18146–18157. Curran Associates, Inc., 2020.
> [10] Kaiyu Yang, Aidan Swope, Alex Gu, Rahul Chalamala, Peiyang Song, Shixing Yu, Saad
> Godil, Ryan J Prenger, and Animashree Anandkumar. Leandojo: Theorem prov-
> ing with retrieval-augmented language models. In A. Oh, T. Naumann, A. Glober-
> son, K. Saenko, M. Hardt, and S. Levine (eds.), Advances in Neural Information Pro-
> cessing Systems, volume 36, pp. 21573–21612. Curran Associates, Inc., 2023.

---

### Official Review · Reviewer_y32e · 2024-11-03

**Soundness:** 3
**Presentation:** 3
**Contribution:** 2
**Rating:** 5
**Confidence:** 4

**Summary:**

This paper presents a framework for generating synthetic Lean theorems and proofs to supplement the training corpus for neural theorem provers. The main approach involves forward proving, where proof states from existing Lean libraries are transformed using a small set of curated proof tactics and model-selected premises. The effectiveness of this framework has been empirically demonstrated on the miniF2F dataset, with the Falcon2-11B model showing improved performance when fine-tuned on the synthetically augmented Lean Workbook dataset.

**Strengths:**

- Well-motived objective: augmenting existing formal corpus synthetically is surely of immerse interest, given the limited amount of formal corpus available.
- The paper is relatively well-written and easy to follow, and with good amount of implementation details in the appendix for reproducibility.

**Weaknesses:**

- There is a lack of experiments validating the effectiveness of using diverse (synthetic) theorems. Based on my experience, synthetic data augmentation typically provides significant benefits, even when theorems are randomly mutated. A key innovation of this paper is the algorithm depicted in Figure 3, which aims to promote diversity. However, in the subsequent experiments, it seems the authors have not established a robust metric to assess the diversity of the generated theorems or to quantify how this diversity impacts the performance of fine-tuned neural provers.
- As an alternative method for augmenting the formal corpus, it would be beneficial to include more comparisons between autoformalization and the proposed synthetic approach. Does the synthetic approach generate more diverse theorems? With an equal amount of additional corpus from each method, would the synthetic approach yield better performance on the fine-tuned prover? Additionally, what impact might combining these two approaches have on overall performance?

**Questions:**

- I would appreciate some more elaboration on why the 'have' tactic needs a lemma from the library to introduce a new hypothesis
- It might be interesting to include a qualitative analysis of the additional theorems proven using the synthetic dataset.

---

> ### Author Response · Authors · 2024-11-25
>
> We thank the reviewer for their thoughtful feedback and suggestions. We respond below to the concerns and questions raised:
>
> > **W1.** …it seems the authors have not established a robust metric to assess the diversity of the generated theorems or to quantify how this diversity impacts the performance of fine-tuned neural provers.
>
> We appreciate your observation of the importance of our search algorithm to our work’s contribution. We did not perform a post-hoc assessment of theorem diversity because there are two relevant but somewhat orthogonal notions of diversity, both of which are principally controlled by the configuration of the generator prior to generating synthetic theorems and for which there does not appear to be consensus on clear evaluation metrics. However, we have added an additional metric regarding subject-matter diversity, which we explain below.
>
> The first relevant notion of diversity is the diversity of the proofs themselves. The algorithm in Figure 3 promotes this type of diversity by ensuring that proofs diverge from one another as early as possible. Because this is the case, there is little room to further promote diversity in the search algorithm itself, since any other algorithm could only produce proofs with greater overlap among them.
>
> The second relevant notion of diversity is the diversity of the subject matter of the theorem statements themselves—i.e., the mathematical fields to which they apply. To promote this form of diversity, we ran our generator over a large number of Mathlib modules, representing a wide array of modern mathematics. Since the generator draws its initial proof states (including the initial hypotheses from which the generator reasons forward) from these modules, the theorems it produces will span many mathematical disciplines. To further quantify this diversity of topics, we have added a table in Appendix H displaying the number of theorems in our synthetic dataset generated using initial states from each top-level Mathlib submodule (e.g., `Algebra`, `Analysis`, `Topology`).
>
> > **W2.** …it would be beneficial to include more comparisons between autoformalization and the proposed synthetic approach.
>
> We appreciate the reviewer’s questions about comparisons between autoformalization and our synthetic approach. We would like to clarify a key point about the relationship between these approaches: Our synthetic method is complementary to and can be applied to any proof dataset, whether human-written or autoformalized. Rather than viewing these as competing approaches that need direct comparison, our method serves as an enhancement that can augment any existing proof corpus.
>
> In our experiments, we demonstrated this versatility by successfully generating synthetic corpora from both human-written proofs (the Mathlib dataset discussed in §5.1 and Tables 1–2) and autoformalized proofs (the Lean Workbook dataset used for the experiments in §5.2 and Table 3). In both cases, fine-tuning on the synthetically expanded datasets led to improved model performance compared to fine-tuning on the original datasets alone.
>
> The empirical results suggest our method provides value regardless of the proof source, making direct comparisons between autoformalization and synthesis less relevant to evaluating our contribution. The more pertinent comparison is between models fine-tuned on a given dataset with and without our synthetic augmentation—a comparison our experiments directly address.

---

> > ### Author Response · Authors · 2024-11-25
> >
> > > **Q1.** I would appreciate some more elaboration on why the 'have' tactic needs a lemma from the library to introduce a new hypothesis.
> >
> > The `have` tactic in Lean requires a lemma from the library to introduce a new hypothesis because the user must show (by supplying such a lemma) that the newly added hypothesis follows from the existing ones. It is possible that this confusion is arising from the use of the word “hypothesis.” In Lean, it is conventional to refer to the collection of locally-known facts in one's context as "hypotheses," regardless of whether they were introduced as an antecedent/assumption of the theorem statement (i.e., the usual informal use of the term "hypothesis") or as a deduced consequence of such assumptions. The hypotheses introduced by the `have` tactic are of the latter form: `have` is used to add to the context a new "locally-true" fact that follows from the existing hypotheses. It does this by applying some existing lemma whose antecedents match the in-context hypotheses. Therefore, the user must specify which lemma they intend to apply to construct the new hypothesis (as well as which in-context hypotheses satisfy its antecedents). We alluded to this terminological detail in passing in §2 of our original submission, but we recognize that this nonstandard usage may be confusing for some readers; we appreciate your highlighting this ambiguity and, consequently, have added an additional clarification to that section.
> >
> > > **Q2.** It might be interesting to include a qualitative analysis of the additional theorems proven using the synthetic dataset.
> >
> > Thank you for this suggestion; we have added an analysis of the additional proved theorems to Appendix G.

---

> > > ### Comment · Reviewer_y32e · 2024-12-02
> > >
> > > Thanks to the authors for their responses—they’ve addressed some of my concerns. That said, I still feel the effectiveness of synthetic theorems hasn’t been fully demonstrated:
> > > - Even with a massive number of synthetic theorems (1 billion vs. 208 million tokens in Mathlib, as pointed out by Reviewer tn5n), the improvement seems relatively small (e.g., just 3 additional theorems).
> > > - While Figure 3 and Appendix H show efforts to promote diversity in synthetic theorem generation—and most of us agree that diverse synthetic theorems can help LLM-based theorem proving—the diversity hypothesis for the Lean proving task doesn’t feel sufficiently tested. There aren’t convincing ablation studies or a clear link between diversity and the prover’s performance.
> > >
> > > Because of this, I’ve decided to stick with my original score.
> > >
> > > On the ``have`` tactic: the authors said it ‘introduces a new hypothesis using a lemma from the library.’ My impression, though, is that it introduces a hypothesis that can then be proved like a normal lemma or theorem, using various tactics and lemmas from the library—not just a single lemma. Either way, thanks for clarifying!

---

### Meta-Review · Area_Chair_6qqr · 2024-12-22

**Metareview:**

This paper proposes a novel framework for generating synthetic theorems and proofs in Lean. New theorems and proofs are generated through forward proving, and a search procedure is enhanced to encourage diversity of theorems and proofs. Experimental evaluation shows modest improvement with the newly synthetic dataset. At a high level, creating a new and diverse dataset for theorem proving is a valuable and important contribution. However, the main concern raised by all reviewers and shared by the AC is whether this general idea of dataset synthesis really helps theorem proving. The improvement of approximately 1% (i.e., 3 problems) is somewhat encouraging, but not convincing enough to show its effectiveness. The authors are encouraged to pursue this line of work, since the general methodology sounds promising, while more careful development and evaluation especially regarding diversity should be conducted convincingly.

**Additional Comments On Reviewer Discussion:**

There are active discussions between authors and reviewers. Relatively minor clarification questions and missing recent related works are carefully addressed by the authors and further acknowledged by reviewers. Careful discussions and new evaluation suggestions regarding the major concern of sufficient effectiveness of the proposed approaches are engaged. However, the current work as it is hasn't yet convinced reviewers as well as the AC.

---

### Decision · Program_Chairs · 2025-01-22

Reject